# The Comprehensive Evaluation of Safflowers in Different Producing Areas by Combined Analysis of Color, Chemical Compounds, and Biological Activity

**DOI:** 10.3390/molecules24183381

**Published:** 2019-09-17

**Authors:** Zong-Jin Pu, Shi-Jun Yue, Gui-Sheng Zhou, Hui Yan, Xu-Qin Shi, Zhen-Hua Zhu, Sheng-Liang Huang, Guo-Ping Peng, Yan-Yan Chen, Ji-Qing Bai, Xiao-Ping Wang, Shu-Lan Su, Yu-Ping Tang, Jin-Ao Duan

**Affiliations:** 1Jiangsu Collaborative Innovation Center of Chinese Medicinal Resources Industrialization, and Jiangsu Key Laboratory for High Technology Research of TCM Formulae, and National and Local Collaborative Engineering Center of Chinese Medicinal Resources Industrialization and Formulae Innovative Medicine, Nanjing University of Chinese Medicine, Nanjing 210023, China; 15951878596@163.com (Z.-J.P.); zhouguisheng1@126.com (G.-S.Z.); glory-yan@163.com (H.Y.); shixuqin@126.com (X.-Q.S.); 18913133908@163.com (Z.-H.Z.); guopingpeng@sohu.com (G.-P.P.); dja@njucm.edu.cn (J.-A.D.); 2Key Laboratory of Shaanxi Administration of Traditional Chinese Medicine for TCM Compatibility, and State Key Laboratory of Research & Development of Characteristic Qin Medicine Resources (Cultivation), and Shaanxi Key Laboratory of Chinese Medicine Fundamentals and New Drugs Research, and Shaanxi Collaborative Innovation Center of Chinese Medicinal Resources Industrialization, Shaanxi University of Chinese Medicine, Shaanxi University of Chinese Medicine, Xi’an 712046, China; chenyanyan59@163.com (Y.-Y.C.); baijiqing323@126.com (J.-Q.B.); wangxiaoping323@126.com (X.-P.W.); 3Jiangsu Rongyu Pharmaceutical Co., Ltd., Huaian 223200, China; hsl_renshou@126.com

**Keywords:** safflower, color, UPLC-MS^2^, chemometrics, quality evaluation

## Abstract

In the present study, a new strategy including the combination of external appearance, chemical detection, and biological analysis was proposed for the comprehensive evaluation of safflowers in different producing areas. Firstly, 40 batches of safflower samples were classified into class I and II based on color measurements and K-means clustering analysis. Secondly, a rapid and sensitive analytical method was developed for simultaneous quantification of 16 chromaticity-related characteristic components (including characteristic components hydroxysafflor yellow A, anhydrosafflor yellow B, safflomin C, and another 13 flavonoid glycosides) in safflowers by ultra-performance liquid chromatography coupled with triple-quadrupole linear ion-trap tandem mass spectrometry (UPLC-QTRAP^®^/MS^2^). The results of the quantification indicate that hydroxysafflor yellow A, anhydrosafflor yellow B, kaempferol, quercetin, and safflomin C had significant differences between the two types of safflower, and class I of safflower had a higher content of hydroxysafflor yellow A, anhydrosafflor yellow B, and safflomin C as the main anti-thrombotic components in safflower. Thirdly, chemometrics methods were employed to illustrate the relationship in multivariate data of color measurements and chromaticity-related characteristic components. As a result, kaempferol-3-*O*-rutinoside and 6-hydroxykaempferol-3-*O*-β-d-glucoside were strongly associated with the color indicators. Finally, anti-thrombotic analysis was used to evaluate activity and verify the suitability of the classification basis of safflower based on the color measurements. It was shown that brighter, redder, yellower, more orange–yellow, and more vivid safflowers divided into class I had a higher content of characteristic components and better anti-thrombotic activity. In summary, the presented strategy has potential for quality evaluation of other flower medicinal materials.

## 1. Introduction

*Carthamus tinctorius* L. (safflower), one of the oldest oil crops and major *Carthamus* species distributed in all of the world, has been applied in cut flowers, dyeing cosmetics, and coloring and flavoring foods [1,2], and safflower is a famous gynecological herbal medicine (Carthami Flos) in China, Korea, Japan, and other countries [3,4]. Modern pharmacological experiments show that safflower possesses a wide range of biological activities, such as dilating coronary arteries, improving myocardial ischemia, and regulating the immune system, as well as anticoagulation, anti-thrombosis, anti-oxidation properties [4,5,6]. In recent years, due to the medicinal value and health care value, more and more researchers have focused on investigating the active compounds, evaluating quality, and developing safflower products.

Recently, the safety and quality evaluation of safflower has been challenged, because it is often the target of artificial coloration and adulteration. Safflowers from different cultivars have different colors [7,8], such as white, yellow, orange, red, etc. The color of the total bloom stage turns from yellow to red gradually. Shades of orange, yellow, and red flowers are most common in early bloom, but post-bloom colors are darker. Additionally, different harvest times of the safflower cause different colors. The color is still yellow at the early harvest time, and the color turns purple and black at the late harvest time [9]. The light intensity can affect the color of the safflower and cause color changes as well [10]. The color of a safflower is usually related to the content of effective components, and it is an important factor to determine the quality of the safflower. Recent research has discovered that safflowers with different colors have great differences in the content of some chemical components. Xu et al. [11] discovered that when the content of hydroxysafflor yellow A (HSYA) is low, the safflower tends to be less red and darker. Tu et al. [12] indicated that the orange and white flower has a high content of HSYA and kaempferol-3-*O*-β-d-glucoside, respectively. Therefore, the color measurement is an important external appearance index of quality evaluation for raw and processed safflower, for determinations of conformity of safflower quality to certain specifications, or for the evaluation of quality changes as a result of safflower processing, storage, or other factors.

Additionally, in China, safflowers have been planted in Xinjiang, Yunnan, Gansu, Sichuan, and other places [13]. It is generally considered that plants grown in different areas have different contents and types of chemical markers, which are usually selected as important factors to study the quality evaluation and genuine producing areas of many functional foods and/or herbs, such as ginkgo seeds [14], *Ginkgo biloba* leaves [15], and *Ziziphus jujube* [16]. Currently, more than 200 compounds have been isolated and identified from safflower, including quinochalones, flavonoids, alkaloids, polyacetylene, and aromatic glucosides, among which quinochalcones and flavonoids are commonly considered as major bioactive and characteristic components associated with the therapeutic effects of safflower. Hence, the characteristic components (or chemical markers) might be a chemically important index to study the quality evaluation and genuine producing areas of safflower.

Nowadays, more and more studies indicate that quality evaluation and genuine producing areas of traditional Chinese medicines (TCM) or Chinese herbal medicine takes place via system engineering, which requires a lot of external appearances, and chemical and biological researches. However, current pharmacopoeias (such as Chinese Pharmacopoeia, U.S. Pharmacopoeia, and European Pharmacopoeia, etc.) are chemical markers-based, rather than bioactivity-oriented. Hence, to achieve the purpose of holistic quality evaluation of safflower, in this work, a new strategy based on a combination of the methods of external appearance, chemical detection, and biological analysis is proposed for the comprehensive analysis and evaluation of safflower samples from different producing areas (Figure 1). First, different safflower samples were classified into different types based on the results of external appearance of color measurements and K-means clustering analysis. Second, chromaticity-related characteristic components were comprehensive analyzed using UPLC-QTRAP^®^/MS^2^ in different types of safflower based on the above external appearance of color measurements. Third, chemometrics methods were employed to illustrate the relationship in multivariate data of color measurements and chromaticity-related characteristic components. Finally, anti-thrombotic analysis was used to evaluate the main biological activities and verify the suitability of the classification basis of different types of safflower based on the color measurements. By these efforts, the external, chemical, and biological evidence was obtained to unveil the quality of the different sources of safflower, which is beneficial to its holistic and more scientific quality control.

## 2. Results and Discussion

### 2.1. Color Measurements and Classification of Safflower

In the beginning, PCA of 40 samples was based on the content of characteristic components, according to the traditional classification method. It was found that the safflower samples from different regions could not be distinguished well. According to the previous literature, there was no direct evidence illustrating that safflower from different regions were different. According to the results, there was no significant difference in the content of characteristic components in safflowers from different regions. However, a significant difference in the content of characteristic components was presented in safflowers from the same region [17]. Additionally, from the previous reports, safflowers from different regions were also not distinguished based on the contents of the primary and secondary metabolites in safflowers and the in vitro bioactivity [18]. As we all know, the quality of safflowers was evaluated by their important appearance of color. The color measurement was selected as an important external appearance index to evaluate the quality of some vegetables and fruits such as tomatoes [19,20], blood oranges [21], blueberry [22], and bread [23]. Therefore, color measurement might be an effective and convenient method to evaluate the quality control of safflower, which has not previously been reported.

K-means clustering aims to partition *n* observations into *k* clusters, in which each observation belongs to the cluster with the nearest mean, serving as a prototype of the cluster [24]. The clustering algorithm refers to dividing a bunch of unlabeled data into different categories through an iterative process. This method ensures that the same type of data have similar features so that each cluster generated is compact inside, but the categories are independent of each other. Thus, safflowers with similar color characteristics would be classified into one category. In the previous analysis, the number of clusters was chosen to be 3, 4, or 5, but there was no significant difference between the groups in color indicators. At the same time, the color of the sample could be observed by the naked eye as orange, orange–red, red, and dark red. In order to ensure that the categories were independent of each other, the number of clusters was determined to be 2. The results of color measurements and K-means clustering analysis are shown in Table 1. Two types of safflower show significant differences in these five indicators. Ranges of L*, a*, b*, C*, and *h*_ab_ values in class I were 39.500~53.473, 20.403~34.543, 41.053~55.563, 50.422~64.749, and 55.055~69.345, respectively. Ranges of L*, a*, b*, C*, and *h*_ab_ values in class II were 30.930~49.697, 19.630~31.780, 25.860~40.527, 36.017~51.216, and 45.889~62.154, respectively. The values of L*, a*, b*, C*, and *h*_ab_ in class I were higher than class II (*p <* 0.01). This suggests that safflowers classified into class I were redder, yellower, brighter, more orange–yellow, and more vivid to the eye than class II. A comparison of class I and class II is shown in Appendix A.

### 2.2. Analysis of Chromaticity-Related Characteristic Components from Safflower

#### 2.2.1. Selection of Chromaticity-Related Characteristic Components from Safflower

From the many previous reports, it has been confirmed that HSYA has a strong correlation with the color in the safflower [11,12]. Additionally, recent researches have shown that quinochalcones and flavonoids have a certain correlation with safflower color [12,25,26]. Therefore, three quinochalcones (HSYA, SC, and ASYB) and 13 flavonoids were selected, including their chromaticity-related characteristic components, for quantitative analysis in the present research.

#### 2.2.2. Optimization of UPLC-QTRAP^®^/MS^2^ Conditions

This study optimized chromatographic conditions such as column, mobile phase, flow rate, and column temperature. The Thermo Scientific Hypersil GOLD (3 × 100 mm, 1.9 µm) column and the Acquity UPLC BEH C_18_ (2.1 × 100 mm, 1.7 μm) column were chosen to investigate the separation of the target compounds. The result indicates that the Acquity UPLC BEH C_18_ (2.1 × 100 mm, 1.7 μm) column could achieve better separation of 16 compounds. The mobile phase of the previous test, including methano–water, acetonitrile–water, methanol–water with 0.1% formic acid, and acetonitrile–water with 0.1% formic acid, were all tested. Acetonitrile had a stronger elution ability than methanol, which shortened the analysis time and obtained a better peak shape; and 0.1% formic acid kept the mobile phase at a suitable pH. The flow rate was set at 0.40 mL/min, and the column temperature was set at 35 °C, which were good for separating target compounds.

To optimize the QTRAP^®^/MS^2^ conditions, Q1 full scans were conducted under both positive and negative electrospray ionization (ESI) modes [27]. The Analyst 1.6 software automatically collected and optimized the optimal ion fragmentation, declustering potential (DP), entrance potential (EP), collision energy (CE), collision cell exit potential (CXP), and other conditions. Three quinochalcone glucosides (HSYA, SC, and ASYB) were more sensitive in the negative ion mode. The MS/MS product ions at *m/z* 491.12 of HSYA resulted from the sugar moieties lost C_4_H_8_O_4_. SC lost C_11_H_10_O_4_ by cleavages of ring A, then lost C_4_H_8_O_4_ and formed the product ion at *m/z* 286.9. ASYB formed the product ion at *m/z* 449.11 (the base peak), together with a complementary product ion at *m/z* 593.15 in low abundance [28]. Flavonol [M + H]^+^ product ions dehydrated to [M + H − H_2_O]^+^, followed by two sequential losses of CO: [M + H − H_2_O − CO]^+^ and [M + H − H_2_O − 2CO]^+^. These losses of carbon monoxide were also observed directly from the protonated flavonoid [M + H − CO]^+^ and [M + H − 2CO]^+^, and the loss of CO and dehydration occurred in the C-ring and proved the flavonol structure of the C-ring of an unidentified flavonoid [15]. Flavones [M + H]^+^ ions exhibited only the [M + H − H_2_O]^+^ and [M + H − H_2_O − CO]^+^ fragments [19]. The optimized mass spectrometry conditions are shown in Table 2, and the chromatograms of the respective compounds are shown in Appendix A.

#### 2.2.3. Method Validation

The established method was verified by measuring the linear, intraday and interday precision, stability, repeatability, and recovery. The calibration functions obtained by plotting the peak area versus the concentration of the compound were linear, and the determination coefficient higher than 0.9935 for all compounds. The limit of detections (LOD) and limit of quantitation (LOQ) ranges of 16 compounds were 1.182~176.130 ng/mL and 8.240~352.260 ng/mL, respectively. Calibration curves, correlation coefficients, linearity ranges, LODs, and LOQs of the 16 compounds are shown in Appendix A. The results of precision, repeatability, stability, and recovery are shown in Appendix A. The results of the experiment indicate that the developed method was accurate and reliable in this study.

#### 2.2.4. Application to the Analysis of Real Samples

The results of quantitative determination are shown in Table 3. The contents of 16 compounds in two categories of safflower were compared, just as in Appendix A. Safflower had a higher content of HSYA, SC, ASYB, kaempferol-3-*O*-rutinoside, 6-hydroxykaempferol-3-*O*-β-d-glucoside, and 6-hydroxykaempferol-3,6,7-tri-*O*-β-d-glucoside in class I. The content of HSYA and ASYB showed great differences between these two types of safflower (*p* < 0.01). HSYA (3.619~19.278 mg/g) and ASYB (2.196~14.124 mg/g) were the two compounds with the highest content. The color of HSYA and ASYB powders was orange–yellow, so the higher content of these two ingredients made the value of b* higher, which further caused the safflower to be more orange–yellow. Kaempferol, quercetin, and SC also had significant differences.

### 2.3. Chemometrics Classification of Safflower Samples from Different Production Areas Based on the Chromaticity-Related Characteristic Components

During recent years, there has been increasing interest in illuminating the intrinsic relationship of multivariate data which are composed of numerous variables measured from many samples by validated analytical methods. Chemometrics methods have been widely used for depicting the intrinsic similarities and differences of samples by comparing particular chemical and biological parameters in different samples. The central idea of unsupervised PCA is to reduce the dimensionality of a data set consisting of a large number of interrelated variables. PCA is a very useful classification technique and widely used in the field of analytical chemistry. OPLS-DA is one of the most popular and common supervised methods. OPLS-DA is increasing its uses related to different chemical issues and could be considered as a popular tool in multivariate analysis. In this study, PCA was performed to classify safflower samples from different production areas based on the contents of the 16 chromaticity-related characteristic components. The supervised method of OPLS-DA was utilized to further separate clusters and validate the classify model. The scatter plots of PCA and OPLS-DA (Figure 2) show that the two types of safflower can be distinguished well. Compounds with VIP values larger than 1 were considered to be more important in the classification than other components. The VIP value represents the contribution of the variable to the model, If the value is larger, the contribution is greater. The VIP values of HSYA, ASYB, kaempferol, quercetin, and SC were more than 1. Thus, these five compounds were considered to be closely related to color characteristics.

More and more researches have pointed out that the color is associated with bioactive compounds content [29]. Pearson correlation matrix analysis was used to discover the correlation between color indicators and compounds’ contents [30]. The results of the correlation analysis are presented in Figure 3. The red and blue part in the figure indicate a strong correlation (*r* > 0.5 or *r* < −0.5). As shown in Figure 3, L* had a strong positive association with HSYA (*r* = 0.597), ASYB (*r* = 0.707), and 6-hydroxykaempferol-3-*O*-β-d-glucoside (*r* = 0.506). Additionally, a* had a strong positive association with HSYA (*r* = 0.597), and a strong negative association with kaempferol (*r* = −0.710) and quercetin (*r* = −0.609). The other parameter of b* had a strong positive association with HSYA (*r* = 0.639) and ASYB (*r* = 0.664). Chroma had a strong positive association with HSYA (*r* = 0.706) and ASYB (*r* = 0.576), and a strong negative association with kaempferol (*r* = −0.621). Hue angle had a strong positive association with ASYB (*r* = 0.651), 6-hydroxykaempferol-3-*O*-β-D-glucoside (*r* = 0.578), SC (*r* = 0.619), and kaempferol-3-*O*-rutinoside (*r* = 0.521). It is reasonable to speculate that if the contents of HSYA and ASYB were higher, the color of safflower would turn lighter and more yellow.

The contents of HSYA, ASYB, and SC were higher in class I, but kaempferol and quercetin had a lower content. Guo et al. found that overexpression of CtCHS1 increased the quinochalcone glucoside accumulation and decreased flavonol aglycone and glycoside contents in safflower, and this study was based on the comparison of the yellow and white line safflower [31]. It indicates that accumulation of quinochalcone glucosides would make safflowers turn yellow and red. HSYA, ASYB, and SC belong to the quinochalcone *C*-glycosides, and the contents of the above three quinochalcone *C*-glycosides were higher in class I, while two flavonol aglycones (kaempferol and quercetin) presented a lower content in class I of safflower. Different expressions of CtCHS1 caused the different content in the five compounds, which might lead to differences in the color of the safflower.

### 2.4. Anti-Thrombotic Evaluation of Two Types of Safflower in Zebrafish

In this study, anti-thrombotic analysis was employed to evaluate the main biological activities and verify the suitability of the classification basis of different types of safflower based on the color measurements.

After being treated with 1.5 μmol/L PHZ for 24 h, thrombus in the caudal vein was obviously increased, but the RBCs decreased significantly (*p* < 0.01), just as shown in Figure 4B, Figure 5B, Figure 6A. The thrombus in the caudal vein was decreased after treatment with aspirin for 24 h, and the RBCs increased significantly (*p* < 0.01), as shown in Figure 4C, Figure 5C, Figure 6A. The thrombus in the caudal vein was decreased and the RBCs increased after treatment with various concentrations of class I and II of safflower; 100 μg/mL of class I of safflower could increase RBCs significantly (*p* < 0.01) and 200 μg/mL of class II of safflower could also increase RBCs significantly (*p* < 0.05). The therapeutic efficacy of aspirin was 41.97%. The therapeutic efficacies of various concentrations of class I of safflower were 18.88%, 19.19%, 33.61%, 41.32%, 46.86%, and 46.80%, and the concentration of EC_50_ was 96.03 μg/mL. The therapeutic efficacies of various concentrations of class II of safflower were 11.94%, 15.14%, 15.26%, 26.92%, 35.48%, and 35.88%, and the concentration of EC_50_ was 181.8 μg/mL.

### 2.5. The Relationship of Color Measurements, Chemical Detection, and Anti-Thrombotic Analysis of Safflower

Safflowers with different colors showed different contents of target compounds. The contents of HSYA, ASYB, kaempferol, quercetin, and SC affected the color of the safflower. HSYA, ASYB, and SC made the safflower brighter, redder, yellower, and more orange–yellow. In addition, 6-hydroxykaempferol-3-*O*-β-d-glucoside and kaempferol-3-*O*-rutinoside made the safflower more orange–yellow too. To compare the activity of two types of safflower, class I of safflower had a better anti-thrombotic activity. The color of class I of safflower was brighter, redder, yellower, more orange–yellow, and more vivid to the eye. These color features of class I of safflower led to the higher content of characteristic components (HSYA, ASYB, and SC) and better anti-thrombotic activity.

Safflower is considered as a medicinal plant for the treatment of cardiovascular diseases, lowering blood cholesterol and relieving the pain. Recently, it was reported that chalcones possess the potential to treat cardiovascular disease [32]. HSYA and ASYB have been confirmed to be the main active chalcones in safflower [33,34,35]. Combined with the results of this study, SC might also be a main active ingredient. These three quinochalcone C-glucosides not only related to the color, but also had an association with anti-thrombotic activity.

Color not only reflected the chemical composition of safflower, but also reflected its pharmacological activity. Safflower which was brighter, redder, yellower, more orange–yellow, and more vivid had a higher content of characteristic quinochalcone C-glucoside components (HSYA, ASYB, and SC) and better anti-thrombotic activity. Therefore, the combination method of color features, chemical detection, and biological analysis is an effective, rapid, and repeatable strategy for the comprehensive analysis and evaluation of different safflower samples.

## 3. Material and Methods

### 3.1. Chemicals and Reagents

Sixteen chemical standards, including hydroxysafflor yellow A (**1**), safflomin C (**2**), anhydrosafflor yellow B (ASYB, **3**), kaempferol (**4**), kaempferol 3-*O*-glucoside (**5**), kaempferol-3-*O*-rutinoside (**6**), kaempferol-3-*O*-β-sophoroside (**7**), 6-hydroxykaempferol (**8**), 6-hydroxykaempferol-3-*O*-β-d-glucoside (**9**), 6-hydroxykaempferol-3,6-di-*O*-β-d-glucoside (**10**), 6-hydroxykaempferol-3,6,7-tri-*O*-β-d-glucoside (**11**), quercetin (**12**), rutin (**13**), luteoloside (**14**), apigenin (**15**), and quercetin-3-*O*-β-d-glucoside (**16**), were used in this study. Compounds **3**, **4**, **7**, **8,** and **10** were purchased from Shanghai Yuanye Biotechnology Company (Shanghai, China). Compounds **5**, **6,** and **13** were bought from Chengdu Chroma-Biotechnology Company (Chengdu, China). Compounds **1**, **12**, **14**, **15,** and **16** were acquired from Nanjing Liangwei Biotechnology Company (Nanjing, China). Compounds **2**, **9,** and **11** were isolated from safflower by the authors [36,37]. Dimethyl sulfoxide (DMSO), phenylhydrazine (PHZ), and acetylsalicylic acid (aspirin) were bought from Aladdin Company (Shanghai, China). *O*-dianisidine was obtained from J&K Scientific Ltd. (Beijing, China). Methanol, acetonitrile, and formic acid were HPLC grade and purchased from Merck (Darmstadt, Germany). Ultrapure water was prepared by the Milli-Q water purification system (Millipore, Bedford, MA, USA).

### 3.2. Plant Materials

Forty batches of samples from different regions were identified as *Carthamus tinctorius* L. by Dr. Hui Yan at the Department of Pharmacognosy, Nanjing University of Chinese Medicine, Nanjing, China. The voucher specimens were deposited at the herbarium in Nanjing University of Chinese Medicine, China. All the samples were pulverized into homogeneous powders (50 mesh), then stored under dry conditions at room temperature. The detailed information of the samples is shown in Table 4.

### 3.3. Color Measurements

The color of the sample powder was measured by Konica Minolta Spectrophotometer CM-5 and each sample was analyzed in triplicates. L* value indicated lightness, from dark (0) to white (100), a lower value indicated darker color and a higher value indicated lighter color. The other CIE parameter of a* value indicated from green (−a*) to red (+a*), while b* value indicated from blue (−b*) to yellow (+b*) [38]. Chroma (*C*∗) was the quantitative colorfulness attribute, as it determined the difference degree in comparison to a grey color with the same lightness for each hue [21]. Hue angle (*h*_ab_) was a parameter that defined the colors traditionally as pinkish, yellowish, and greenish [21]. High values of hue angle represented more reddish–orange color and low values indicated more reddish–blue color. CIE LAB color space is shown in Appendix A. The equations for calculating C*_ab_ and hue angle (*h*_ab_) are Equations (1)–(3) [38]:(1)$$\mathrm{Chroma}(C{*}_{\mathrm{ab}})\;=\;\sqrt{\left({a*}^{2}+{b*}^{2}\right)}$$
(2)$$\mathrm{Hue}\;\mathrm{angle}\;({h{}^{\circ}}_{\mathrm{ab}})\;=\;\tan^{-1}(b*/a*)\;\mathrm{when}\;a*\geq 0\;\mathrm{and}\;b*\;\geq \;0$$
(3)$$\mathrm{Hue}\;\mathrm{angle}\;({h{}^{\circ}}_{\mathrm{ab}})\;=\;180\;+\;\tan^{-1}(b*/a*)\;\mathrm{when}\;a*\;<\;0$$

### 3.4. Quantitative Determination of 16 Compounds

#### 3.4.1. Preparation of Sample Solution

Firstly, 0.5 g of each dry sample powder was weighed accurately into a 50 mL conical flask with stopper, and 20 mL of 50% methanol was added accurately to each conical flask. Then, after weighing the total weight, ultrasonic extraction (40 kHz) for 40 min at 30 °C was conducted, using the same solvent to replenish the loss weight during extraction. After centrifugation (13,000× *g*, 10 min), the supernatants were stored at 4 °C and filtered through 0.22 μm cellulose membrane filters prior to injection.

#### 3.4.2. Preparation of Standard Solution

A mixed standard stock solution containing the above target compounds **1**–**16** was dissolved by 70% methanol. The initial concentration of the compounds **1**–**16** were 875.625 μg/mL, 54.375 μg/mL, 64.375 μg/mL, 32.5 μg/mL, 40.234 μg/mL, 38.734 μg/mL, 81.875 μg/mL, 58.750 μg/mL, 63.750 μg/mL, 60 μg/mL, 33.437 μg/mL, 62.5 μg/mL, 63.125 μg/mL, 9.609 μg/mL, 21.562 μg/mL, and 45.089 μg/mL, respectively. Then, the mixed standard stock solution was diluted with 70% methanol to a series of appropriate concentrations [39]. All the standard solutions were stored at 4 °C until use, and filtered through a 0.22 μm cellulose membrane before injection.

#### 3.4.3. UPLC-QTRAP^®^/MS^2^ Conditions

Chromatographic separation was performed on a Waters ACQUITY UPLC system (Waters, Milford, MA, USA). An UPLC BEH C_18_ column (2.1 × 100 mm, 1.7 µm) was used for analyzing all the samples. The mobile phase included A (0.1% formic acid aqueous solution) and B (acetonitrile) (*v*/*v*). The gradient elution procedure was as follows: 10–20% B at 0–2 min, 20–30% B at 2–7 min, 30–80% B at 7–12 min, 80–100% B at 12–13 min, and 100–10% B at 13–14 min. The mobile phase was set at a flow rate of 0.40 mL/min, and the injection volume was 2 μL. The column temperature was set at 35 °C and the sample temperature was 4 °C.

Mass spectrometry detection was performed with an AB SCIEX Triple Quad 6500 plus (AB SCIEX Corp., Massachusetts, USA) equipped with an electrospray ionization source (ESI). The ESI–MS were acquired in both positive and negative ion multiple reaction monitoring (MRM) modes, with the capillary voltage of 5 kV, the desolvation gas flow rate set to 1000 L/h at a temperature of 550 °C, the cone gas flow rate set at 50 L/h, and the source temperature at 150 °C. The cone voltage (CV) and collision energy (CE) [40] were set to match the MRM of each compound. The dwell time was automatically set by the MultiQuant software 3.0.2 (AB SCIEX Corp., Massachusetts, USA).

#### 3.4.4. Validation of UPLC-QTRAP^®^/MS^2^ Method

A series of concentrations of standard solutions was prepared for obtaining the calibration curve. The limit of detections (LODs) and the limit of quantitations (LOQs) of 16 compounds were acquired while the signal-to-noise ratios (S/N) were 3 and 10, respectively. The peak height divided by the background noise value was calculated as the S/N.

The 16 standard solutions with six replicates were investigated during a single day and three consecutive days, respectively, for the intra-day and inter-day precision, and the results were displayed as the RSDs of the peak area for each standard compounds. The repeatability was confirmed by six different sample solutions, which were prepared from the same sample **S26**. Then, the variations were expressed by RSDs. Meanwhile, the sample solutions mentioned above were stored at 4 °C, injected, and analyzed at 0, 2, 4, 6, 8, and 16 h for the evaluation of the stability.

The purpose of the recovery test was to evaluate the accuracy of the developed method. It was performed by adding the 16 standards at low (80% of known amounts), medium (same as known amounts), and high (120% of known amounts) levels into the representative sample **S26** solution. The spiked samples were then extracted, processed, and quantified in accordance with the methods mentioned above. The spike recoveries were calculated using Equation (4):Recovery = [(measured amount − original amount)/spiked amount] × 100%(4)

### 3.5. Anti-Thrombotic of Two Types of Safflower in Zebrafish

Firstly, 1 mL of the sample solution was accurately prepared according to the above steps. Then, the solvent was evaporated and residue was dissolved in 25 mL of 0.1% DMSO medium [41]. The final concentration of the solution was 1000 μg/mL, then diluted with water as required. **S28** and **S29** were selected as representative samples, which both showed different color and represented two types of safflower.

Zebrafish at 48 HPF (hours post fertilization) were bought from Nanjing EzeRinka Biotechnology Co., Ltd (Nanjing, China). Firstly, 20 zebrafish larvae were placed in a 24-well microplate and treated with 1.5 μmol/L PHZ for 24 h [42]. Secondly, PHZ was washed out, and then different groups of zebrafish were respectively treated with the positive drug of aspirin (5.625 μg/mL) and two types of safflower at different dosing concentrations (25, 50, 100, 200, 400, and 600 μg/mL). These different doses were consistent with the experimental concentrations, and the concentration of 50% maximal effect (EC_50_) was calculated at the same time. Larvae placed in 0.1% DMSO were considered to be the vehicle control. The morphology of zebrafish was evaluated after incubation for 24 h at 28 °C. Thirdly, 10 zebrafish of each group were stained with 1.0 mg/mL *O*-dianisidine dye liquor for 15 min and washed with DMSO three times [43]. Lastly, the specimens were placed on glass slides and observed by a fluorescent inverted microscope (Leica, Germany). The heart red blood cells (RBCs) were quantitatively analyzed by Image-Pro Plus 6.0, based on the staining intensity (SI) of erythrocytes in the heart. The anti-thrombotic effects of safflower and aspirin were evaluated calculated by the following formula [42] Equation (5).
(5)$$\mathrm{Therapeutic}\;\mathrm{efficacy}\;(\%)\;=\;\frac{\mathrm{SI}(\mathrm{drug})\;-\;\mathrm{SI}(\mathrm{model})}{\mathrm{SI}(\mathrm{control})\;-\;\mathrm{SI}(\mathrm{model})}\times 100\%$$

### 3.6. Statistical Analysis

Data of color measurements were imported to SPSS 24.0 software (SPSS, Chicago, USA) for K-Means clustering analysis. The data were expressed as mean ± SEM, and two-tailed *t*-test and analysis of variance (ANOVA) were performed by GraphPad Prism 7 (San Diego, CA, USA). A value of *p* < 0.05 was regarded as a significant difference, and a value of *p* < 0.01 was regarded as a very significant difference. Principal components analysis (PCA) and orthogonal partial least squares discriminant analysis (OPLS-DA) was performed by SIMCA-P 14.0 software (Umetrics, Umea, Sweden). Correlation analysis and heat-map were performed by Origin 2017 software (OriginLab, Massachusetts, USA).

## 4. Conclusions

HSYA had the highest content in safflower, and it was the main active ingredient [44,45]. Studies have also pointed out that HSYA content was related to the color of safflower. This research indicates that the content of ASYB and SC can also affect the color and its anti-thrombotic activity. The higher content of HSYA, ASYB, and SC makes safflower redder, yellower, brighter, more orange–yellow, and more vivid, due to the strong correlation between the color and compounds content. Furthermore, the high content of these compounds leads to better anti-thrombotic activity. Therefore, safflowers with different colors might have different contents of compounds and different pharmacological activities. Moreover, it is also feasible to evaluate the quality of other flower medicinal materials based on this strategy.

## Figures and Tables

**Figure 1 molecules-24-03381-f001:**
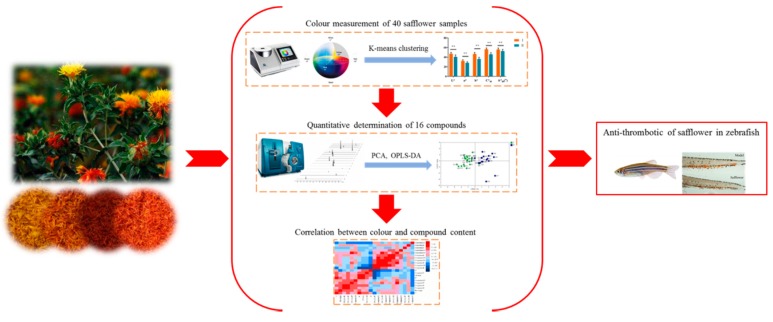
Technology roadmap of this study.

**Figure 2 molecules-24-03381-f002:**
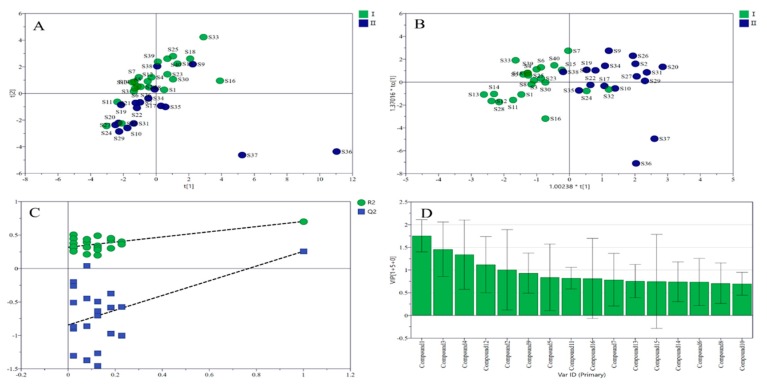
PCA (**A**), OPLS-DA (**B**), permutation test (**C**), and VIP value (**D**) based on the content of the 16 compounds in two categories.

**Figure 3 molecules-24-03381-f003:**
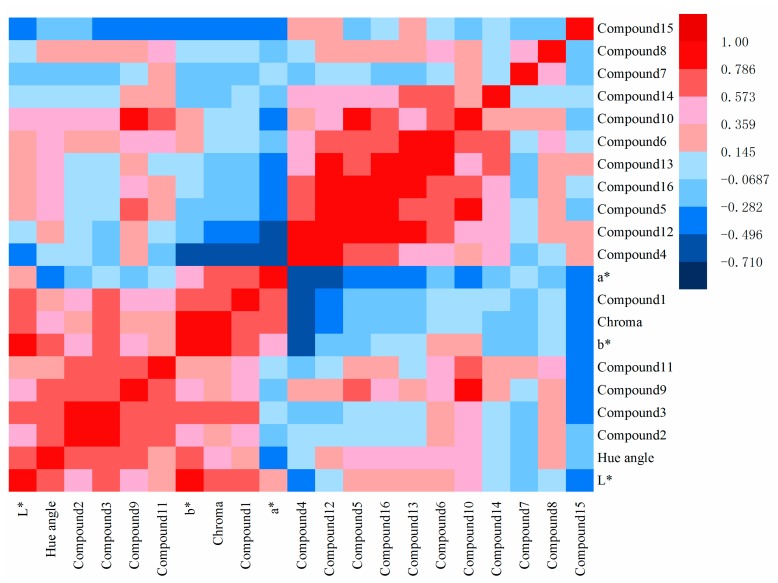
Correlation between color indicators and compounds content.

**Figure 4 molecules-24-03381-f004:**
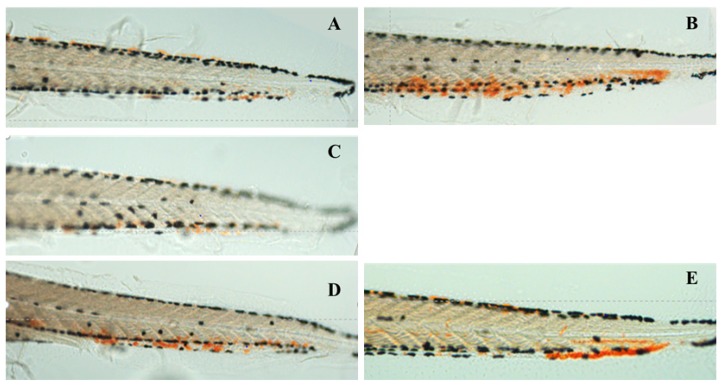
Thrombus in the caudal vein of vehicle control group (**A**), model group (**B**), aspirin group (**C**), the group treated with 200 μg/mL class I of safflower (**D**), and the group treated with 200 μg/mL class II of safflower (**E**).

**Figure 5 molecules-24-03381-f005:**
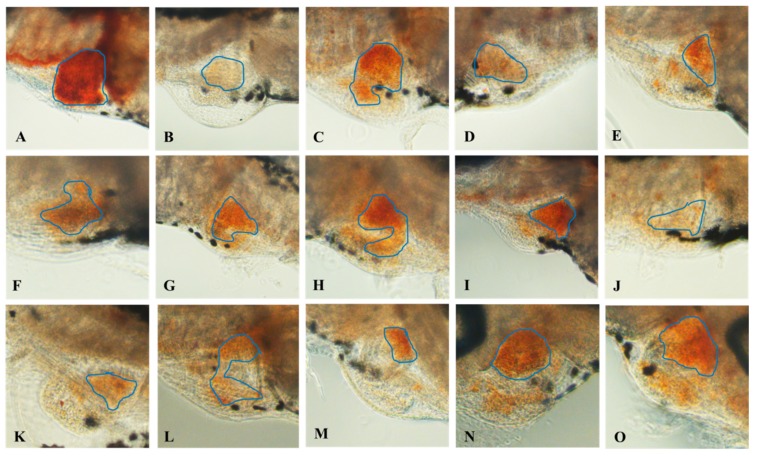
The heart red blood cells (RBCs) of the control group (**A**), model group (**B**), aspirin group (**C**), various concentrations of class I of safflower (25–600 μg/mL) (**D**–**I**), and various concentrations of class II of safflower (25–600 μg/mL) (**J**–**O**). The blue circle indicates the staining of erythrocytes in the heart.

**Figure 6 molecules-24-03381-f006:**
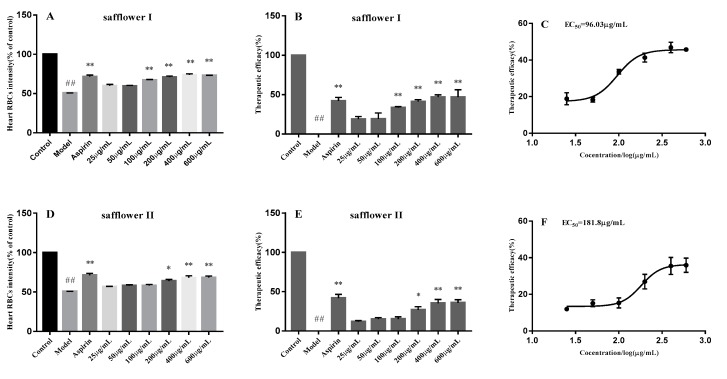
The heart RBCs, the therapeutic efficacies, and the concentration of EC_50_ in class I of safflower (**A**–**C**) and class II of safflower (**D**–**F**). Data are expressed as mean ± SEM of three separate experiments (*n* = 6). ^##^
*p* < 0.001 versus vehicle control group; ** *p* < 0.01 and *** *p* < 0.001 versus model group.

**Table 1 molecules-24-03381-t001:** Color measurements and classification of safflower.

No	L*	a*	b*	Chroma	Hue Angle	Classification	No	L*	a*	b*	Chroma	Hue Angle	Classification
**S1**	47.3	33.3	47.7	58.2	55.1	I	**S21**	33.6	28.4	35.5	45.5	51.4	II
**S2**	38.3	29.0	35.1	45.5	50.4	II	**S22**	42.0	29.6	36.5	47.0	51.0	II
**S3**	46.7	33.0	43.3	54.5	52.7	I	**S23**	47.5	34.3	47.0	58.1	53.8	I
**S4**	46.8	31.8	43.8	54.1	54.0	I	**S24**	39.7	32.1	44.6	55.0	54.2	I
**S5**	48.1	33.2	55.6	64.7	59.1	I	**S25**	47.3	32.4	46.5	56.7	55.1	I
**S6**	48.3	31.6	47.4	56.9	56.3	I	**S26**	37.0	29.4	40.5	50.1	54.1	II
**S7**	43.2	31.9	42.3	53.0	53.0	I	**S27**	36.0	27.4	29.0	39.8	46.6	II
**S8**	46.1	31.4	43.1	53.3	53.9	I	**S28**	49.2	34.5	48.5	59.5	54.5	I
**S9**	44.6	31.1	38.6	49.5	51.1	II	**S29**	36.3	27.8	32.0	42.4	49.1	II
**S10**	38.5	29.3	38.2	48.1	52.5	II	**S30**	45.7	32.7	43.5	54.5	53.1	I
**S11**	48.9	33.3	48.9	59.2	55.7	I	**S31**	30.9	25.1	25.9	36.0	45.9	II
**S12**	48.9	33.5	48.4	58.8	55.3	I	**S32**	39.5	32.5	42.7	53.7	52.7	I
**S13**	48.9	32.7	46.7	57.0	55.0	I	**S33**	53.5	20.4	54.1	57.8	69.3	I
**S14**	48.8	33.6	48.0	58.5	55.0	I	**S34**	42.5	31.8	40.2	51.2	51.6	II
**S15**	46.8	32.7	44.9	55.6	53.9	I	**S35**	42.7	26.7	36.3	45.1	53.6	II
**S16**	40.4	31.1	42.5	52.6	53.9	I	**S36**	43.4	22.4	37.2	43.5	58.9	II
**S17**	40.4	29.6	38.0	48.1	52.1	II	**S37**	49.7	19.6	37.2	42.0	62.2	II
**S18**	50.9	26.3	49.3	55.9	61.9	I	**S38**	44.2	29.0	39.9	49.3	54.0	II
**S19**	43.2	27.5	37.6	46.5	53.8	II	**S39**	43.0	29.9	41.1	50.8	53.9	I
**S20**	35.9	27.0	29.0	39.6	47.0	II	**S40**	45.1	28.8	41.4	50.4	55.1	I

**Table 2 molecules-24-03381-t002:** The ion mode and parameters for MRM of 16 compounds.

	Compound	t_R_ (min)	Ion mode	MRM Transition	DP	EP	CE	CXP
**1**	hydroxysafflor yellow A	3.04	[M − H]^−^	611.06 > 490.8	−105	−10	−36	−53
**2**	safflomin C	12.20	[M − H]^−^	613.125 > 286.9	−40	−10	−45	−10
**3**	anhydrosafflor yellow B	8.80	[M − H]^−^	1043.122 > 448.9	−5	−10	−40	−10
**4**	kaempferol	12.99	[M − H]^−^	284.89 > 117	−155	−10	−56	−13
**5**	kaempferol-3-*O*-glucoside	8.86	[M − H]^−^	447.109 > 283.8	−120	−10	−32	−10
**6**	kaempferol-3-*O*-rutinoside	8.40	[M − H]^−^	593.003 > 284.9	−120	−10	−44	−29
**7**	kaempferol-3-*O*-β-sophoroside	5.78	[M + H]^+^	611.8 > 286.9	56	10	31	20
**8**	6-hydroxykaempferol	12.31	[M + H]^+^	303.933 > 168.8	186	10	47	14
**9**	6-hydroxykaempferol-3-*O*-β-d-glucoside	5.66	[M + H]^+^	466.271 > 304.1	66	10	21	32
**10**	6-hydroxykaempferol-3,6-di-*O*-β-d-glucoside	4.68	[M + Na]^+^	649.805 > 347	106	10	45	34
**11**	6-hydroxykaempferol-3,6,7-tri-*O*-β-d-glucoside	2.70	[M + H]^+^	789.891 > 303.9	61	10	41	34
**12**	quercetin	12.56	[M − H]^−^	300.859 > 150.8	−90	−10	−28	−17
**13**	rutin	6.41	[M − H]^−^	609.048 > 300	−150	−10	−52	−33
**14**	luteoloside	7.22	[M − H]^−^	447.045 > 284.8	−80	−10	−36	−31
**15**	apigenin	12.95	[M + H]^+^	270.989 > 153.1	171	10	49	14
**16**	quercetin-3-*O*-β-d-glucoside	6.88	[M − H]^−^	461.98 3 > 298.9	−115	−10	−32	−31

**Table 3 molecules-24-03381-t003:** The content of 16 compounds in 40 batches of safflower (mg/g).

No	Compound Number															
	1	2	3	4	5	6	7	8	9	10	11	12	13	14	15	16
**S1**	12.542	0.423	8.601	0.172	0.072	0.475	0.013	0.000	0.309	0.208	1.014	0.125	0.183	0.006	0.035	0.140
**S2**	7.279	0.225	2.975	0.166	0.070	0.437	0.019	0.000	0.384	0.219	1.095	0.049	0.065	0.009	0.037	0.044
**S3**	14.583	0.277	4.351	0.138	0.061	0.501	0.013	0.098	0.332	0.159	0.701	0.034	0.074	0.006	0.034	0.041
**S4**	15.525	0.336	6.179	0.149	0.080	0.629	0.018	0.091	0.372	0.234	0.998	0.039	0.080	0.005	0.034	0.030
**S5**	11.456	0.265	4.989	0.142	0.055	0.628	0.020	0.101	0.335	0.135	0.669	0.029	0.100	0.005	0.000	0.039
**S6**	12.316	0.239	4.482	0.129	0.065	0.536	0.023	0.097	0.361	0.180	0.826	0.023	0.068	0.006	0.032	0.017
**S7**	11.803	0.256	4.361	0.151	0.065	0.503	0.025	0.099	0.411	0.180	0.984	0.028	0.056	0.004	0.000	0.025
**S8**	13.968	0.286	5.310	0.141	0.085	0.605	0.017	0.000	0.375	0.226	0.998	0.045	0.116	0.006	0.034	0.053
**S9**	13.611	0.240	5.812	0.165	0.259	0.580	0.023	0.096	0.576	0.444	1.378	0.059	0.074	0.006	0.000	0.112
**S10**	8.548	0.183	2.575	0.247	0.060	0.429	0.011	0.000	0.215	0.119	0.599	0.097	0.090	0.006	0.037	0.038
**S11**	15.204	0.188	5.437	0.147	0.061	0.450	0.004	0.000	0.263	0.128	0.504	0.019	0.080	0.004	0.035	0.048
**S12**	19.278	0.279	7.429	0.165	0.085	0.605	0.007	0.000	0.391	0.152	0.908	0.025	0.119	0.006	0.033	0.086
**S13**	17.615	0.245	7.250	0.131	0.073	0.527	0.008	0.000	0.382	0.169	0.756	0.012	0.089	0.004	0.033	0.042
**S14**	15.912	0.283	6.935	0.151	0.079	0.568	0.006	0.000	0.332	0.135	0.661	0.019	0.101	0.005	0.000	0.064
**S15**	17.714	0.363	7.098	0.222	0.202	0.454	0.010	0.000	0.629	0.315	1.152	0.048	0.060	0.007	0.000	0.136
**S16**	18.580	0.303	6.679	0.197	0.143	1.070	0.018	0.091	0.485	0.181	1.480	0.074	0.208	0.033	0.035	0.132
**S17**	11.124	0.225	3.561	0.185	0.090	0.536	0.015	0.096	0.310	0.220	0.939	0.095	0.143	0.013	0.033	0.080
**S18**	13.795	0.582	10.157	0.180	0.121	0.631	0.010	0.093	0.555	0.330	1.170	0.059	0.111	0.007	0.033	0.122
**S19**	8.689	0.200	3.439	0.142	0.061	0.472	0.015	0.095	0.270	0.137	0.672	0.031	0.059	0.002	0.033	0.033
**S20**	6.605	0.171	2.215	0.236	0.056	0.357	0.011	0.096	0.195	0.093	0.664	0.051	0.049	0.004	0.036	0.023
**S21**	8.745	0.329	3.913	0.198	0.074	0.562	0.014	0.097	0.323	0.111	0.759	0.046	0.059	0.004	0.039	0.023
**S22**	10.158	0.273	4.006	0.202	0.062	0.626	0.009	0.091	0.244	0.112	0.709	0.049	0.086	0.006	0.037	0.034
**S23**	15.730	0.343	7.145	0.190	0.098	0.617	0.010	0.093	0.562	0.241	1.025	0.052	0.102	0.005	0.033	0.062
**S24**	8.232	0.150	2.559	0.184	0.038	0.438	0.007	0.000	0.158	0.094	0.397	0.038	0.061	0.004	0.036	0.023
**S25**	17.862	0.377	8.845	0.171	0.101	0.643	0.009	0.099	0.493	0.264	1.176	0.040	0.114	0.005	0.000	0.094
**S26**	8.295	0.299	4.204	0.169	0.089	0.574	0.019	0.095	0.421	0.226	1.229	0.050	0.075	0.005	0.037	0.037
**S27**	7.625	0.159	2.332	0.232	0.056	0.353	0.012	0.000	0.198	0.111	0.655	0.051	0.051	0.005	0.037	0.032
**S28**	17.452	0.238	7.022	0.162	0.074	0.562	0.004	0.000	0.442	0.153	0.745	0.028	0.104	0.007	0.032	0.049
**S29**	5.806	0.178	2.333	0.252	0.051	0.430	0.009	0.000	0.218	0.109	0.592	0.067	0.051	0.004	0.044	0.031
**S30**	15.171	0.306	7.633	0.200	0.111	0.752	0.015	0.097	0.446	0.199	1.162	0.065	0.122	0.004	0.033	0.099
**S31**	7.495	0.304	2.746	0.292	0.088	0.360	0.013	0.000	0.267	0.117	0.755	0.109	0.061	0.004	0.041	0.045
**S32**	7.678	0.175	2.196	0.188	0.056	0.465	0.010	0.000	0.238	0.118	0.572	0.043	0.063	0.012	0.039	0.020
**S33**	13.284	0.757	14.124	0.227	0.126	0.707	0.008	0.096	0.738	0.314	1.351	0.055	0.083	0.003	0.032	0.090
**S34**	9.598	0.288	4.077	0.207	0.079	0.631	0.019	0.094	0.327	0.184	0.982	0.053	0.078	0.003	0.036	0.039
**S35**	9.160	0.308	4.245	0.185	0.083	0.568	0.015	0.095	0.352	0.227	0.789	0.083	0.161	0.010	0.039	0.121
**S36**	5.855	0.301	4.242	0.394	0.488	1.157	0.013	0.094	0.698	0.504	0.907	0.417	0.352	0.024	0.039	0.609
**S37**	3.619	0.186	3.642	0.331	0.318	0.906	0.007	0.094	0.390	0.285	0.681	0.307	0.301	0.012	0.041	0.296
**S38**	14.954	0.370	7.559	0.182	0.124	0.500	0.009	0.000	0.517	0.185	1.146	0.028	0.057	0.011	0.000	0.071
**S39**	16.308	0.359	8.346	0.182	0.104	0.567	0.011	0.095	0.504	0.164	1.045	0.021	0.073	0.007	0.000	0.054
**S40**	16.456	0.441	9.403	0.180	0.135	0.513	0.011	0.092	0.561	0.198	1.276	0.032	0.062	0.011	0.033	0.011

**Table 4 molecules-24-03381-t004:** Forty batches of safflower samples from different regions.

No	Location	No	Location
**S1**	Hefei City, Anhui Province	**S21**	Huocheng County, Xinjiang Uygur Autonomous Region
**S2**	Jiuquan City, Gansu Province	**S22**	Jimsar County, Xinjiang Uygur Autonomous Region
**S3**	Jinhe Village, Jiuquan City, Gansu Province	**S23**	Jimsar County, Xinjiang Uygur Autonomous Region
**S4**	Jinhe Village, Jiuquan City, Gansu Province	**S24**	Jimsar County, Xinjiang Uygur Autonomous Region
**S5**	Nanqu Village, Yumen City, Gansu Province	**S25**	Jimsar County, Xinjiang Uygur Autonomous Region
**S6**	Nanqu Village, Yumen City, Gansu Province	**S26**	Jimsar County, Xinjiang Uygur Autonomous Region
**S7**	Nanqu Village, Yumen City, Gansu Province	**S27**	Emin County, Tacheng City, Xinjiang Uygur Autonomous Region
**S8**	Dongqu Village, Yumen City, Gansu Province	**S28**	Emin County, Tacheng City, Xinjiang Uygur Autonomous Region
**S9**	Yumen county, Yumen City, Gansu Province	**S29**	Yumin County, Tacheng City, Xinjiang Uygur Autonomous Region
**S10**	Luoyang City, Henan Province	**S30**	Urumqi City, Xinjiang Uygur Autonomous Region
**S11**	Qixia District, Nanjing City, Jiangsu Province	**S31**	Huocheng County, Xinjiang Uygur Autonomous Region
**S12**	Qixia District, Nanjing City, Jiangsu Province	**S32**	Yining County, Xinjiang Uygur Autonomous Region
**S13**	Qixia District, Nanjing City, Jiangsu Province	**S33**	Yutian County, Xinjiang Uygur Autonomous Region
**S14**	Qixia District, Nanjing City, Jiangsu Province	**S34**	Emin County, Tacheng City, Xinjiang Uygur Autonomous Region
**S15**	Yongxin Couty, Jian City, Jiangxi Province	**S35**	Lijiang City, Yunnan Province
**S16**	Jianyang City, Sichuan Province	**S36**	Yongsheng County, Yunnan Province
**S17**	Xichang City, Sichuan Province	**S37**	Yongsheng County, Yunnan Province
**S18**	Chabuchar County, Xinjiang Uygur Autonomous Region	**S38**	Suzhou District, Jiuquan City, Gansu Province
**S19**	Yutian County, Xinjiang Uygur Autonomous Region	**S39**	Suzhou District, Jiuquan City, Gansu Province
**S20**	Huocheng County, Xinjiang Uygur Autonomous Region	**S40**	Suzhou District, Jiuquan City, Gansu Province

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
