# Peer review of "The Comprehensive Evaluation of Safflowers in Different Producing Areas by Combined Analysis of Color, Chemical Compounds, and Biological Activity"

_molecules, 2019, doi:10.3390/molecules24183381_

Round 1

Reviewer 1 Report

I have reviewed an article: "The comprehensive evaluation of safflower in different producing areas by combinating colour, chemical components and biological activity analysis". Authors are dealing with evaluation of safflower according to their appearance, chemical composition and biological activity. Article is well written, all conclusions are supported by experimental data and below are listed issues that might improve a quality of article;

1) Abstract section: please indicate what components are characteristic for safflower and which of them are suspected to have anti-thrombotic effect

2) Material and methods section: Please explain why did you tested 25-600 ug/mL of safflower extracts for antithrombotic effect? Are these concentrations relevant to those that are already used? 

Author Response

1. Abstract section: please indicate what components are characteristic for safflower and which of them are suspected to have anti-thrombotic effect.

Reply: Thanks for your comment. Done. The relative contents were added in Abstract section, please see lines 8, 9, 15 and 16 in page 2.

2. Material and methods section: Please explain why did you tested 25-600 ug/mL of safflower extracts for antithrombotic effect? Are these concentrations relevant to those that are already used?

Reply: Thanks for your comment. In order to calculate the EC50 for anti-thrombotic activity, we tried a series of dosing concentrations. When the concentrations was greater than 600 ug/mL, the survival rate of zebrafish was significantly reduced, so the highest concentration was chosen 600 ug/mL. These series of concentrations were consistent with the experimental concentrations. For details, please see lines 15-17 in page 13.

Reviewer 2 Report

The manuscript “The comprehensive evaluation of safflower in different producing areas by combinating colour, chemical components and biological activity analysis” is devoted to the actual problem of analytical and medicinal chemistry. The reviewed article is interesting and theme of the article meets the scope of the journal. Work is performed at sufficient scientific level and has good quality; the results of investigation are professionally interpreted.

This manuscript can be accepted for publication. My decision is accept.

Author Response

This manuscript can be accepted for publication. My decision is accept.

Reply: Thanks for your comment.

Reviewer 3 Report

To be frank, I am pretty impressed with Your valuable manuscript (MS). Well done, indeed. My sincere congrats to all authors. Overall Merit: HIGH

Accept after minor revision.

The English language requires improvement. Please, kindly improve it as much as possible.

In addition to this, the authors are kindly requested to consider citing of the following references (related to a TCHM remedy for heart disease including hypertension):

- Natural Product Research 2012, Volume 26, Issue 8,        Pages 696-702                                                         - Natural Product Research 2012, Volume 26, Issue 3,        Pages 209-215                                                          - Cryptogamie Bryologie 2011, Volume 32, Issue 2,            Pages 113-117 

Without a doubt, Your promising MS has a real potential to be quite well cited (in terms of its hetero-citations) in the time to come. Fortunately, TCHM has yet much to offer.                                    

Last but not least, very best of (research) luck ahead to You all.      

Author Response

1. The English language requires improvement. Please, kindly improve it as much as possible.

Reply: Thanks for your comment. We have tried our best to improve English writing, please see the red words and sentences in the revised manuscript.

2. In addition to this, the authors are kindly requested to consider citing of the following references (related to a TCHM remedy for heart disease including hypertension):

- Natural Product Research 2012, Volume 26, Issue 8, Pages 696-702.

Natural Product Research 2012, Volume 26, Issue 3, Pages 209-215.

Cryptogamie Bryologie 2011, Volume 32, Issue 2, Pages 113-117.

Reply: Thanks for your comment. The three conferences was added in the revised manuscript, please see pages 27 and 28.

3. Without a doubt, Your promising MS has a real potential to be quite well cited (in terms of its hetero-citations) in the time to come. Fortunately, TCHM has yet much to offer. Last but not least, very best of (research) luck ahead to You all.

Reply: Thanks for your comment.